# The Effect of Statin on Anemia in Patients with Chronic Kidney Disease and End-Stage Kidney Disease: A Systematic Review and Meta-Analysis

**DOI:** 10.3390/jpm12071175

**Published:** 2022-07-19

**Authors:** Meng-Hsu Tsai, Fu-You Su, Hao-Yun Chang, Po-Cheng Su, Li-Yun Chiu, Michal Nowicki, Chih-Chin Kao, Yen-Chung Lin

**Affiliations:** 1Division of Nephrology, Department of Internal Medicine, School of Medicine, College of Medicine, Taipei Medical University, 250, Wu-xing St., Taipei 110, Taiwan; b101104009@tmu.edu.tw (M.-H.T.); sueandjeff22000@yahoo.com.tw (F.-Y.S.); b101105041@tmu.edu.tw (H.-Y.C.); b101105148@tmu.edu.tw (P.-C.S.); b101105017@tmu.edu.tw (L.-Y.C.); salmonkao@gmail.com (C.-C.K.); 2Department of Medical Education, Taipei Medical University Hospital, Taipei 110, Taiwan; 3Division of Nephrology, Department of Internal Medicine, Taipei Medical University Hospital, 252, Wu-xing St., Taipei 110, Taiwan; 4Department of Nephrology, Hypertension and Kidney Transplantation, Central University Hospital, Medical University of Lodz, 92-213 Lodz, Poland; lp.zdol.demu@ikciwon.lahcim; 5TMU Research Center of Urology and Kidney (TMU-RCUK), Taipei Medical University, Taipei 110, Taiwan

**Keywords:** anemia, anti-inflammation, meta-analysis, statin, ESKD

## Abstract

Although erythropoietin-stimulating agents are effective in treating anemia in patients with end-stage kidney disease (ESKD) undergoing hemodialysis, some ESKD patients, especially those with inflammation, continue to suffer from anemia. Statin, an inhibitor of hydroxymethylglutaryl-CoA (HMG-CoA) reductase with lipid-lowering effects, may have a pleiotropic effect in reducing inflammation, and thus increase hemoglobin (Hb) level. We searched the PubMed, Embase, and Cochrane databases for relevant studies. The population of interest comprised advanced chronic kidney disease (CKD) patients and ESKD patients receiving hemodialysis with statin treatment. The included study designs were randomized control trial/cohort study/pre-post observational study, and outcomes of interest were Hb, erythropoietin resistance index (ERI) and ferritin. PRISMA 2020 guidelines were followed, and risk of bias (RoB) was assessed using the RoB 2.0 tool in randomized controlled trials, and the Newcastle-Ottawa scale (NOS) in cohort studies. We eventually included ten studies (5258 participants), comprising three randomized controlled trials and seven cohort studies. Overall, Hb increased by 0.84 g/dL (95% confidence interval [CI]: −0.02 to 1.70) in all groups using statins, including single-arm cohorts, and by 0.72 g/dL (95% CI: −0.02 to 1.46) in studies with placebo control. Hb levels were higher in the study group than in the control group, with a mean difference of 0.18 g/dL (95% CI: 0.04–0.32) at baseline and 1.0 g/dL (95% CI: 0.13–1.87) at the endpoint. Ferritin increased by 9.97 ng/mL (95% CI: −5.36 to 25.29) in the study group and decreased by 34.01 ng/mL (95% CI: −148.16 to 80.14) in the control group; ferritin fluctuation was higher in the control group. In conclusion, statin may improve renal anemia in ESKD patients receiving hemodialysis and regular erythropoietin-stimulating agents. Future studies with more rigorous methodology and larger sample size study should be performed to confirm this beneficial effect.

## 1. Introduction

Anemia is a frequent consequence of advanced chronic kidney disease (CKD), and the prevalence is even higher in patients with end-stage kidney disease (ESKD) [1,2,3]. In addition, anemia in CKD is associated with increased risk of cardiovascular disease, in-hospital mortality, and reduced quality of life [4]. The use of erythropoietin-stimulating agents (ESAs) in patients with ESKD has decreased the requirement of blood transfusions, thereby reducing the transfusion-associated complications [5,6] and improving patients’ quality of life. However, the management of anemia in patients with CKD has encountered several difficulties, particularly resistance to ESA therapy. An estimated 5%–10% of patients with renal anemia have insufficient response to ESA, which is defined as an inadequate increase in the hemoglobin (Hb) level, despite standard or even higher doses of ESA [7]. Moreover, patients with ESA resistance require more erythropoietin to reach the target Hb level, which results in increased risk of cardiovascular diseases, mortality, and costs [8].

To overcome this challenge, the mechanisms underlying renal anemia have been extensively studied. Multifactorial causes were identified to explain the anemia in patients with CKD, including relative erythropoietin deficiency, uremic-induced inhibition of erythropoiesis, shortened erythrocyte survival, inflammatory state of patients, and disordered iron homeostasis [3,9]. Furthermore, resistance to ESAs may be positively correlated with the inflammatory status of patients or iron status in patients who underwent ESA treatment [9,10]. Therefore, treating anemia with a combination of agents targeting multiple etiologies may be a solution. In general, the erythropoietin resistance index (ERI) and Hb are classical markers of the treatment effect on renal anemia.

Statin has been found to exert pleiotropic effects, such as anti-inflammation, antifibrosis, antioxidation, and endothelial function improvement in patients with kidney failure [11,12,13,14,15]. Non–lipid-lowering application of statin in patients with CKD and ESKD is gradually being explored. The anti-inflammatory effect of statin may improve the chronic inflammatory state in patients with kidney failure, which is related to renal anemia and ESA resistance [9]. However, the result of previous studies has had limited effect in reaching consensus on statin supplements.

Hence, we performed a meta-analysis to evaluate the effect of statins in patients with kidney failure, especially those receiving hemodialysis. We explored whether the usage of statin should be considered as a treatment in patients with CKD and ESKD with anemia.

## 2. Materials and Methods

### 2.1. Database, Search Terms, and Strategies

This systematic review and meta-analysis followed the preferred reporting items for systematic reviews and meta-analyses (PRISMA) 2020 statement (listed as Appendix A) [16]. We searched the PubMed, Embase, and Cochrane Library databases for studies published in English before July 2021. The key words/MeSH terms and our search strategies used are presented in the Appendix A (listed as Appendix A). We also manually searched the reference lists of the included studies to identify additional relevant articles. In addition, ClinicalTrials.gov was searched for registered trials that had been completed but not yet published. Two authors (SP Cheng and LY Chiu) independently searched the databases and identified eligible studies based on their titles and abstracts, followed by full-text review. Any discrepancies were resolved through discussion and consensus. Final results were reviewed by a senior investigator (YC Lin).

### 2.2. Selection Criteria

Studies were included if they met the following criteria: (1) participants with CKD or ESKD, (2) treatment arms with statin, (3) randomized controlled trial (RCT), cohort study, or pre-post observational study and (4) primary or secondary outcomes similar to our outcomes of interest (Hemoglobin, Erythropoietin resistance index, Ferritin). In addition, articles were excluded if they met the following criteria: (1) participants with early stage CKD (stage I-II), (2) case-control/case series study.

### 2.3. Data Extraction and Quality Assessment

Two authors (MH Tsai and HY Chang) independently reviewed the included studies and extracted relevant data, including study design, patient characteristics (CKD stage, sex), type and dosage of statin, follow-up duration, and outcomes of interest (Hb, ERI, and ferritin). The data of our primary outcomes were extracted as means ± standard deviations (SDs). One study [KOC 2011] represented its ferritin data with median and interquartile range. Hence, we converted the data format to mean and standard deviation [17]. We calculated the ERI of one study [Sirken 2003] by using the formula [18]: (weekly erythropoietin/weight)/Hb(IU/kg/week/g/dL)

To determine the missing SD of the change from baseline, we calculated the correlation coefficient based on the formula in the Cochrane handbook Ch 16.1.3.2. The actual change-from-baseline data that we needed were provided by one study [Masajtis 2018]. Next, the missing SD of the change from baseline of other studies was imputed using the correlation coefficient. The SD was transformed to the standard error (SE) by using the formula in the Cochrane handbook because the SE is compatible with R software. (Analysis data is available in Appendix A) The bias of the included studies was assessed using two tools: the Cochrane Collaboration risk of bias tool 2.0 [19] for RCTs and the Newcastle–Ottawa scale (NOS) for cohort studies; an NOS score of ≥7 was considered high-quality.

### 2.4. Statistical Analysis

Meta-analyses were performed using R software (version 3.6.3) with random effects model [20,21,22,23,24,25,26,27,28,29]. “Meta” and “metafor” packages were used to perform the meta-analysis. The effect of intervention was primarily based on comparisons between the experimental groups of the included studies, because the study design was mostly single-arm without a control group. We used mean differences and their 95% confidence intervals (CIs) to express continuous outcomes (changes from baseline to follow-up). Hb and ferritin were analyzed using the single-arm meta-analysis R code. ERI would not be analyzed in the absence of the actual change-from-baseline data. Statistic heterogeneity was measured using the *I*^2^ test, and publication bias was tested using Egger’s test.

## 3. Results

### 3.1. Study Selection

Figure 1 presents the flowchart of study screening and selection. The initial search strategy yielded 3380 articles, and 2003 remained after duplicates were removed. By reviewing the titles, we excluded 1878 studies and one not written in English. Abstract review led to the removal of another 98 articles, including those that were not RCTs or observational cohort studies. The full texts of the 26 remaining studies were retrieved, and, of them, 13 did not include statin as intervention. In addition, a manual search of the reference lists of these studies led to the inclusion of one additional study. Finally, 10 eligible studies were included for this meta-analysis.

### 3.2. Study and Patient Characteristics

We included 10 studies (3 RCTs and 7 cohort studies), with a total of 5258 participants. Table 1 summarizes the study and patient characteristics. Not all the included studies provided baseline data on Hb, ERI, and ferritin: eight studies provided data on Hb; nine of them provided data on ferritin, and three studies provided data on ERI (Table 1). Among the 10 studies, 9 of 10 included atorvastatin as their intervention with the most common doses of 20 mg/day. Most of the studies (8/10) were conducted on ESKD patients who received regular hemodialysis, and the remaining 2 studies were conducted on advanced stage CKD patients. In addition, the follow-up length of all studies ranged from 4 weeks to more than 2 years.

### 3.3. Quality Assessment

We assessed the methodological quality of each study on the basis of predefined criteria. First, we assessed randomized controlled trials with the risks of bias tool 2.0 (Figure 2 and Figure 3). Two studies revealed probable bias of the randomization process, and one study showed probable bias due to deviations from intended intervention and high risk of bias due to missing outcome data. Among the 3 RCTs, the overall risks of bias were uneven (Low risk: Some concerns: High risk = 1:1:1). Second, the Newcastle-Ottawa scale (NOS) was utilized to assess the quality of the non-randomized studies (Figure 4). Comparability and selection of non-exposed cohort could not be assessed in 3 studies due to their lacking a non-exposed group, which made the full marks of these 3 studies reduce from 9 to 6. In addition, the total score of these 3 studies were 3,4, and 4 respectively, and the score of the remaining studies ranged from 4–8.

### 3.4. Outcomes

#### 3.4.1. Hemoglobin

First, we assessed the mean difference of Hb at baseline and endpoint (Figure 5 and Figure 6). The mean difference of Hb levels between the experimental and control groups was 0.18 g/dL (95% CI: 0.04–0.32 g/dL) at baseline and 1.00 g/dL (95% CI: 0.13–1.87 g/dL) at the endpoint. Heterogeneity assessed with the I^2^ value at baseline and at the endpoint was 0% and 91%, respectively. In spite of the heterogeneity, endpoint values were high, and the result indicated that the use of statin contributed to an increase in Hb level.

Second, the single-arm forest plot of Hb, with a total sample size of 194, was represented (Figure 7). The pooled estimate of the mean difference of Hb was 0.84 g/dL (95% CI: −0.02 to 1.70 g/dL), and the I^2^ value was 100%. Although the difference was not statistically significant, we discovered an incremental tendency in the Hb level that was consistent with the results of baseline and endpoint between-groups analysis.

Third, we assessed the effect of statin on patient’s Hb level by analyzing the mean difference of change-from-baseline data between statin group and control group (Figure 8). The values did not differ significantly (MD: 0.72 g/dL; 95% CI: −0.02 to 1.46 g/dL) between the 2 groups, with a total sample size of 1520. In addition, the I^2^ value for the Hb assessment was 99%, which indicated high heterogeneity across studies. Despite the high heterogeneity and lack of statistically significant results, we observed that statin use slightly increased Hb levels and reduced the levels of serum ferritin, which is a known inflammatory marker.

#### 3.4.2. Ferritin

The mean difference in ferritin levels between the experimental and control groups was 9.97 ng/mL (95% CI: −5.36 to 25.29 ng/mL) at baseline (Figure 9) and −34.01 ng/mL (95% CI: −148.16 to 80.14 ng/mL) at the endpoint (Figure 10). Heterogeneity assessed with the I^2^ value at baseline and endpoint was 0% and 68%, respectively. This indicated the decremental tendency of ferritin after the use of statin, even though it was statistically nonsignificant.

#### 3.4.3. Erythropoietin Resistance Index

Due to insufficient data to conduct statistical analysis, we performed a systematic review of the effect of statin on ERI. Three studies mentioned the change of ERI after statin use. Among the three studies, two of them solely presented the change of ERI of statin group. In the study by Nand et al., the ERI of the statin group decreased significantly from 38.67 ± 11.33 IU/kg/week/g/dL to 26.81 ± 5.71 IU/kg/week/g/dL, whereas the change of ERI of the control group did not differ significantly (baseline: 43.62 ± 12.18 IU/kg/week/g/dL, endpoint: 39.67 ± 9.78 IU/kg/week/g/dL, *p* value = 0.336). In Tsouchnikas et al., the ERI of the statin group decreased significantly from 8.34 ± 3.70 IU/kg/week/g/dL to 7.87 ± 3.10 IU/kg/week/g/dL. Likewise, the result of Sirken et al. showed a decreased tendency of ERI (baseline: 10.63 ± 7.62 IU/kg/week/g/dL, endpoint: 6.72 ± 4.77 IU/kg/week/g/dL) after being treated with statin, but the result was statistically nonsignificant. To sum up, the results of three studies consistently indicated that statin might improve resistance to EPO in the patients studied.

### 3.5. Publication Bias

Funnel plots were not generated in our study due to inadequate sample size. Tests for funnel plot asymmetry should include at least 10 studies to provide sufficient statistical power to distinguish chance from real asymmetry.

## 4. Discussion

Our study is the first systematic review and meta-analysis to discuss the effect of statins on ESA hypo-responsiveness, or resistance, in patients with CKD. We focused on whether statin use can ameliorate anemia by measuring the changes in patients’ Hb levels after treatment with statin. Ferritin, a biomarker of the body’s iron storage, is also an acute-phase protein that is upregulated during inflammation [30]. By evaluating the change in serum ferritin after statin treatment, we attempted to determine whether statins exert an anti-inflammatory effect in these patients and whether this effect was correlated with lowered resistance to ESAs. Furthermore, we included both head-to-head and single-arm studies in our analysis to thoroughly evaluate the effect of statins.

The Hb results were consistent in both head-to-head and single-arm studies concerning the effect of statin on anemia, demonstrating both between-group and within-group trends of increased Hb levels after statin use. Moreover, ferritin levels tended to non-significantly decrease after statin use, indicating a decrease in the inflammatory index. These data corresponded to our hypothesis that statins exert anti-inflammatory effects (increased Hb and decreased ferritin levels) in patients with CKD. Inflammation is an essential factor associated with Hb variability [9]. Current evidence indicates that the suppression of bone marrow erythropoiesis and erythropoietin production by the proinflammatory cytokines (IL-1, IL-6, TNF-α) may be the main causes of the inflammatory anemic effect. However, the underlying mechanisms may be complex, and they remain unclear. In addition, higher ferritin level increases C-reactive protein level in patients on hemodialysis, which could be partially explained by the disturbance of iron released from ferritin in patients with inflammation-associated anemia [31]. Hence, increased Hb and decreased ferritin levels are indicators of the anti-inflammatory effect of statins.

ERI is an indicator of patients’ response to ESAs. It is calculated as the weekly average erythropoietin dose per kilogram of body weight per average Hb over 3 months. Due to its simplicity and usefulness, most ESA association studies have used the ERI to represent the degree of resistance to erythropoietin. Although we had insufficient data on ERI to conduct a statistical analysis, data from individual studies provided information on ERI changes with statin use. In Nand et al. [20], ERI decreased significantly in the statin group, whereas its decrease was nonsignificant in the control group. Similarly, Tsouchnikas et al. [25] demonstrated that ERI significantly decreased from 8.34 ± 3.70 IU/kg/week/g/dL to 7.87 ± 3.10 IU/kg/week/g/dL after statin use. Sirken et al. [27] also indicated a decrease in ERI from 10.63 ± 7.62 IU/kg/week/g/dL at baseline to 6.72 ± 4.77 IU/kg/week/g/dL at the study endpoint; however, they did not specify if this change was significant. Taken together, the consistent decreases in ERI imply that statins lower resistance to erythropoietin.

In our analysis, we noticed enormous heterogeneity in the Hb and ferritin data. High heterogeneity was demonstrated in the analysis of baseline and endpoint values and both single-arm and head-to-head comparisons of Hb. Among the included studies, the Hb values presented in Nand et al. [20] were considerably different from those presented in other studies. However, we could not determine the source of heterogeneity because the authors neither tabulated the demographic data of the experimental and control groups nor discussed whether the two groups were comparable. In addition, the data of studies that were included in single-arm and head-to-head analysis were scattered, which may have contributed to high heterogeneity. We conducted leave-one-out analysis to explore the influence of single study on heterogeneity. The incremental tendency of Hb after statin use remained the same after leaving Nand et al. out in the analysis of baseline and endpoint values (Appendix A). Meanwhile, the I^2^ value declined from 91% to 54%. Yet, heterogeneity of single-arm and head-to-head comparison were still considerably high after the analysis (Appendix A). Similarly, marked heterogeneity was noted in the analysis of the endpoint value of serum ferritin (I^2^ = 68%). The difference between the two groups shown in Nand et al. was much greater than that in other studies, but we could not determine the reason underlying the differences. We concluded that the high heterogeneity in our results may have occurred because a large proportion of our studies were observational studies, even though we failed to identify the source of heterogeneity. Future large-scale RCTs are required to clarify this issue.

Our study had several limitations. First, most of the included studies had uneven and relatively small sample sizes. Second, most of our included studies were observational studies (7/10, 70%). In addition, the observational studies were single-arm studies. Since RCTs are considered the most powerful study type for estimating effects, the small number of RCTs in our study may have weakened the strength of our meta-analyzed results. Third, not all of the included studies were pooled in all analyses because some studies lacked the outcome of interest. For example, the study by Hasegawa et al. [22] met our search criteria, but their data on patients’ Hb and ferritin were unfeasible, making the study impracticable for inclusion in analysis. Fourth, our pooled estimates of Hb and ferritin may be different from the clinical conditions because too few trials provided sufficient data for analysis. For instance, only two studies measured ferritin levels in the control group. Thus, these results may not apply to the whole population. Last, but not least, the follow-up duration of included studies was diverse, which ranged from 12 weeks to 56 months. One study pointed out that the effect of statin was much less effective with an insignificant reduction of the effect with treatment duration (>3 months) in patients receiving hemodialysis [32]. In other words, the beneficial effect of statins was only observed in short therapy duration (<3 months) and the effect may decline as the therapy duration became longer. Since the follow-up period of our included studies were at least 3 months, we may have underestimated the effect of statin.

## 5. Conclusions

In conclusion, the results of this meta-analysis revealed that statin therapy in patients with CKD caused a trend of increased Hb and decreased ferritin levels. However, our results were not statistically significant, and we could not analyze ERI because of insufficient data. Future large-scale, well-designed, prospective, randomized trials are required to validate our results.

## Figures and Tables

**Figure 1 jpm-12-01175-f001:**
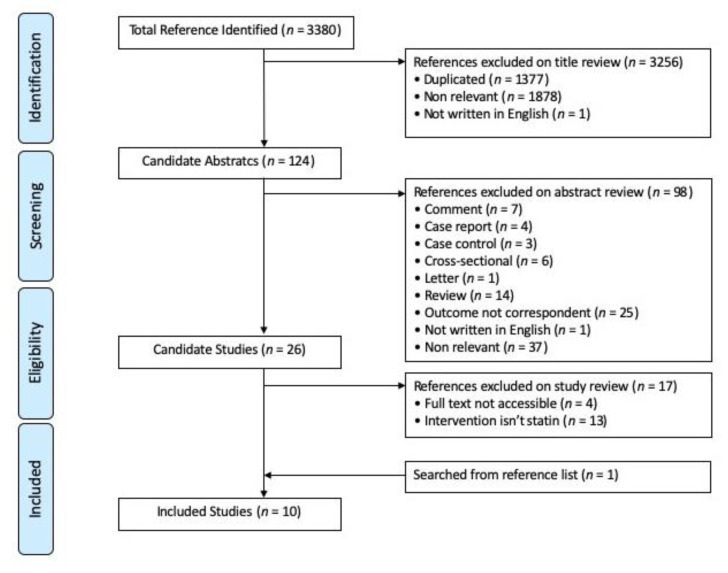
PRISMA flow chart.

**Figure 2 jpm-12-01175-f002:**
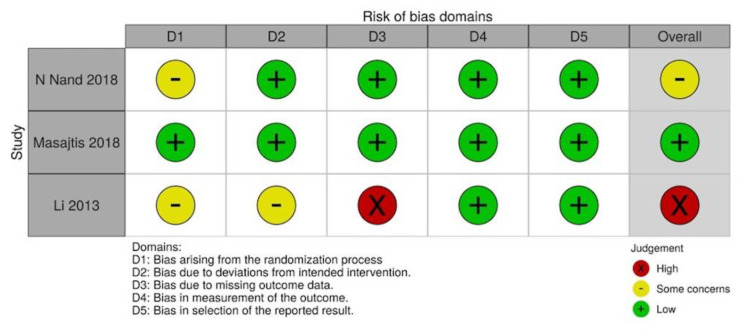
Risk of bias of randomized controlled trials.

**Figure 3 jpm-12-01175-f003:**
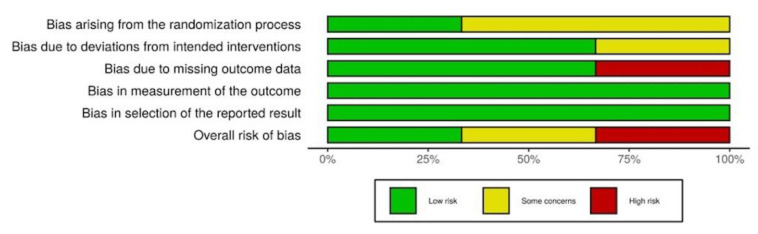
Risk of bias of randomized controlled trials.

**Figure 4 jpm-12-01175-f004:**
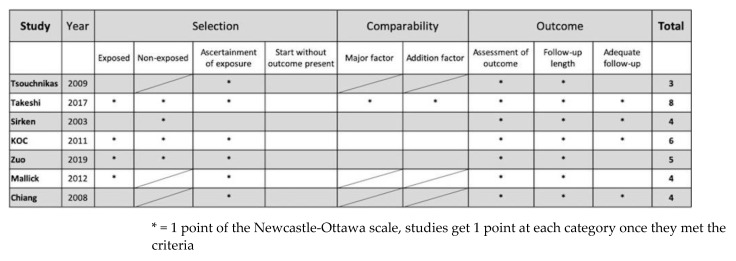
The Newcastle-Ottawa scale (NOS) of cohort studies.

**Figure 5 jpm-12-01175-f005:**
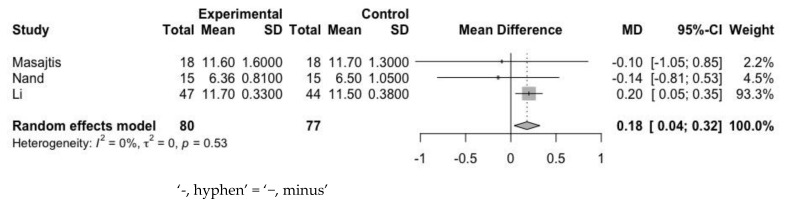
Mean difference of Hb between experimental group and control group at baseline.

**Figure 6 jpm-12-01175-f006:**
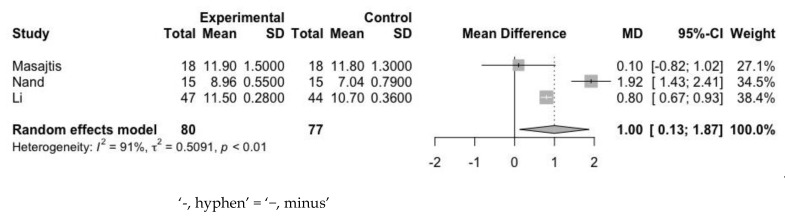
Mean difference of Hb between experimental group and control group at endpoint.

**Figure 7 jpm-12-01175-f007:**
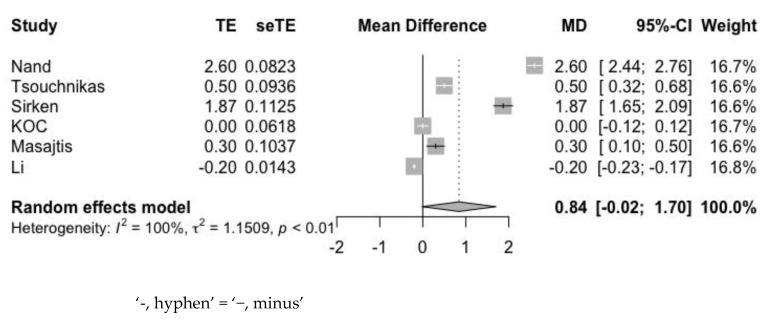
Mean difference of Hb between baseline and endpoint of study group solely.

**Figure 8 jpm-12-01175-f008:**
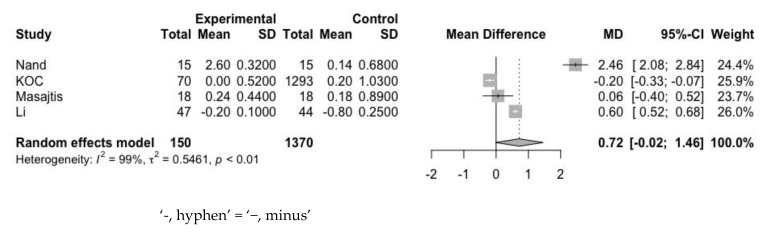
Mean difference of change-from-baseline value of Hb between statin group and control group.

**Figure 9 jpm-12-01175-f009:**
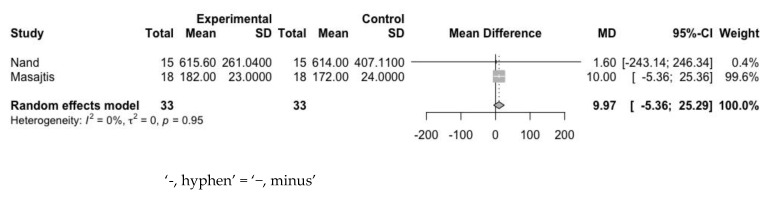
Mean difference of ferritin between experimental group and control group at baseline.

**Figure 10 jpm-12-01175-f010:**
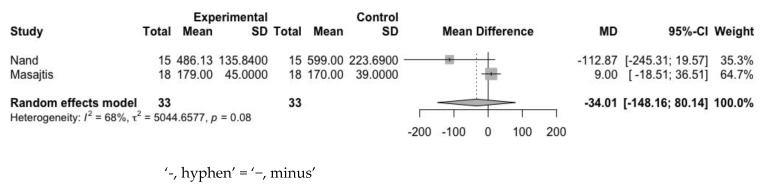
Mean difference of ferritin between experimental group and control group at endpoint.

**Table 1 jpm-12-01175-t001:** Summaries of included papers in the study.

Study	N (Total N = 5258)		Statin (Dose)	Baseline Hb (g/dL)		Baseline ERI	Baseline Ferritin (ng/mL)	Patient Characteristics		Follow-Up	Study Type
								Sex (M:F)	CKD Stage		
N Nand 2018	N = 30		Atorvastatin (20 mg/d)	A ^1^	6.36 ± 0.81	38.67 ± 11.33	615.60 ± 261.04	NR	ESKD (HD)	4 months	RCT
	A ^1^	B ^2^									
				B ^2^	6.50 ± 1.05	43.62 ± 12.18	614.00 ± 407.11				
	15	15									
Masajtis 2018	N = 36		Atorvastatin (20 mg)	A ^1^	11.6 ± 1.6	NR	182 ± 23	17:1	CKD stage III/IV	15 months	RCT
	A ^1^	B ^2^									
				B ^2^	11.7 ± 1.3		172 ± 24				
	18	18									
Li 2013	N = 91		Atorvastatin (20 mg/d)	A ^1^	11.7 ± 3.3	NR	NR	26:21	ESKD (HD)	6 months	RCT
	A ^1^	B ^2^									
				B ^2^	11.5 ± 3.8	NR	NR	24:20			
	47	44									
Chiang 2008	N = 30		Atorvastatin (10 mg/d)	NR		NR	641.8 ± 223	12:18	ESKD (HD)	12 weeks	Prospective cohort (Single arm)
Tsouchnikas 2009	N = 25		Atorvastatin (20 mg/d) (40 mg/d) ^4^	12.1 ± 1.1		8.34 ± 3.70	443.3± 203.3	14:11	ESKD (HD)	9 months	Prospective cohort (Single arm)
Takeshi 2017	N = 3602		Atorvastatin Fluvastatin Lovastatin Pravastatin Rosuvastatin Simvastatin	A ^1^	NR	NR	87.0 (37.0–185.0) ^3^	304:281	ESKD (HD)	4 months	Prospective cohort
	A ^1^	B ^2^		B ^2^	NR	NR	99.5 (41.9–213.0) ^3^	1901:1116			
	585	3017									
Sirken 2003	N = 38		Atorvastatin (mean = 18.1 mg) Simvastatin (mean = 24 mg) Cerivastatin (mean = 0.4 mg) Lovastatin (mean = 20 mg) Pravastatin (mean = 20 mg)	A ^1^	10.61 ± 1.2	NR	618 ± 334.1	9:10	ESKD (HD)	4.7 months (mean)	Retrospective cohort
	A ^1^	B ^2^									
	19	19		B ^2^	11.64 ± 0.98	NR	470.2 ± 287	13:6			
KOC 2011	N = 1363		NR	A ^1^	11.1 ± 1.4	NR	625 (388–761) ^3^	35:35	ESKD (HD)	NR	Retrospective cohort
	A ^1^	B ^2^									
				B ^2^	10.8 ± 1.6	NR	612 (337–1000) ^3^	737:556			
	70	1293									
Zuo 2019	N = 200		Atorvastatin (20 mg, N = 35) Atorvastatin (10 mg, N = 11) Rosuvastatin (10 mg, N = 7) Simvastatin (20 mg, N = 6) Simvastatin (40 mg, N = 11)	A ^1^	7.9 ± 1.4	NR	231.1 (89.8–411.6) ^3^	34:43	CKD stage III-V	23.6 ± 13.4 months (6–56 month)	Retrospective cohort
				B ^2^	7.7 ± 1.7	NR	235.3 (81.1–453.7) ^3^	48:75			
	A ^1^	B ^2^									
	77	123									
Mallick 2012	N = 1305		NR	11.8 ± 0.95		Mean 15 ± 14.08 Male 13.5 ± 13.2 Female 17.0 ± 14.8	509.6 ± 228.19	704:601	ESKD (HD)	2 years	Retrospective cohort (Single arm)

CKD: chronic kidney disease, ESRD: end stage renal disease, ESKD: end stage kidney disease, HD: hemodialysis, RCT: randomized controlled trial, NR: not relevant, M:F = male:female. Groups were divided into: ^1^ = Statin prescribed, ^2^ = Without statin prescribed; Single arm studies contain group of using statins solely. ^4^ = tapering up to 40 mg/day if patients’ LDL target wasn’t reached. Values were presented as mean ± SD, except for “ ^3^ ” that are expressed as median (lower and upper quartile).

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
