# Peer review of "The Effect of Statin on Anemia in Patients with Chronic Kidney Disease and End-Stage Kidney Disease: A Systematic Review and Meta-Analysis"

_jpm, 2022, doi:10.3390/jpm12071175_

Round 1

Reviewer 1 Report

Thank you very much for giving me a chance to review the manuscript. 

You may find my suggestion below;

Title

-          The title is explanatory and eye-catching.

Abstract

-          Abbreviations should not be used in the abstract

-          Line 18: What did the authors mean with ‘uremia patients’?

-          Otherwise, the abstract is well-constructed.

- Keywords should be in accordance with the MeSH system.

Introduction

-          Line 45: Grammar check for ‘CKD is crucial to improving their clinical outcomes.

-          Line 40-56: In the first paragraph, the authors presented well-known information about anemia in CKD, the readers who will read the metaanalysis have already known lots about it. I suggest shortening this part.

-          Line 64-70: Statins are generally known drug and basic information about the drug is unnecessary.

-          I suggest re-arranging the introduction about the relationship between anemia in CKD and statins.

Methods

-          Line 92: Why did the authors search for CKD or ESKD? In the title, they said that the metanalysis is about patients undergoing hemodialysis. And also in the exclusion criteria, the authors exclude only the patients with early-stage CKD. The information is contradictory to the Title.  

-          Otherwise, the methods section is well-established.

Results:

-          Line: 133: Two studies were done with advanced-stage CKD patients, I suggest avoiding these studies, or changing the title and methods. But it would be better not to include these two studies and make statistical analysis again.

-          There are lots of data and data associated figures. I suggest giving some written data in the manuscript and some of them as a Figure, not written. It is not readable in this style.

-          Ferritin is not a good marker for patients undergoing hemodialysis, these patients regularly get iron therapy.

Discussion:

-          Line 228-229: ‘. Furthermore, resistance to ESAs may be positively correlated with the inflammatory status of patients or iron status in these patients who underwent ESA treatment [26, 27].’ Should be in the Introduction part.

-          Line 223: As I mentioned in the results, ferritin is not a good marker. ‘By evaluating the change in serum ferritin after statin treatment, we attempted to determine whether statins exert an anti-inflammatory effect in these patients and whether this effect was correlated with lowered resistance to ESAs.

-          Line 229: ‘Moreover, ferritin levels tended to non-significantly decrease after statin use, indicating a decrease in the inflammatory index’ the sentence should be rechecked that I mentioned above.

-          Line 238: Should be rechecked for the same reason about the ferritin.

-          Heterogeneity is the most important part of the study and the authors explained it very well.

Conclusion:

-          The conclusion part is well-written. 

Author Response

Response to Comments

Reviewer 1

Abstract

  1. Abbreviations should not be used in the abstract
  2. Line 18: What did the authors mean with ‘uremia patients’?

Response: Thank you for your valuable reminder. We have corrected all the abbreviations in the abstract. ‘Uremia patients’ in the abstract means those patients who suffered from end-stage kidney disease. We have revised the term ‘Uremia patients’ with ‘ESKD patients’.

Introduction

  1. Line 45: Grammar check for ‘CKD is crucial to improving their clinical outcomes.
  2. Line 40-56: In the first paragraph, the authors presented well-known information about anemia in CKD, the readers who will read the metaanalysis have already known lots about it. I suggest shortening this part.
  3. Line 64-70: Statins are generally known drug and basic information about the drug is unnecessary.
  4. I suggest re-arranging the introduction about the relationship between anemia in CKD and statins.

Response: Thank you for your valuable comments. We have simplified and re-arranged the introduction.

Methods

  1. Line 92: Why did the authors search for CKD or ESKD? In the title, they said that the metanalysis is about patients undergoing hemodialysis. And also in the exclusion criteria, the authors exclude only the patients with early-stage CKD. The information is contradictory to the Title.

Response: Thank you for your valuable remarks. Initially, our research aimed to focus on patient’s disease stage, which is ESKD undergoing hemodialysis. However, to thoroughly investigate in the effect of statin on EPO resistance, we subsequently expanded the study patient group from those in ESKD receiving hemodialysis to advanced-stage CKD and ESKD patients. The title we provided at first was the wrong version, and we have revised it to the latest version. We are very sorry for our careless mistake and the misleading information.

Results

  1. Line: 133: Two studies were done with advanced-stage CKD patients, I suggest avoiding these studies, or changing the title and methods. But it would be better not to include these two studies and make statistical analysis again.

Response: Thank you for your insightful advice. As we mentioned above, we wanted to expand the study patient group for a thorough investigation into the effect of statin on resistance to EPO. Therefore, we included these two studies with advanced-stage CKD patients in our analysis. We have changed the title and method.

  1. There are lots of data and data associated figures. I suggest giving some written data in the manuscript and some of them as a Figure, not written. It is not readable in this style.

Response: Thank you for your valuable remarks. We have revised the description of our data and the sequence of data associated figures.

  1. Ferritin is not a good marker for patients undergoing hemodialysis, these patients regularly get iron therapy.

Response: Thank you for your valuable reminder. Indeed, ferritin is not a good marker of inflammation for patients undergoing hemodialysis due to iron therapy. However, CKD patients do not receive iron supplements as regularly as those who undergoing hemodialysis. Since we included both CKD and ESKD patients, the concern of representative of ferritin as an inflammatory marker may lowered.

Discussions

  1. Line 228-229: ‘. Furthermore, resistance to ESAs may be positively correlated with the inflammatory status of patients or iron status in these patients who underwent ESA treatment [26, 27].’ Should be in the Introduction part.

Response: Thank you for your reminder. We have re-arranged the sentence to the introduction part.

  1. Line 223: As I mentioned in the results, ferritin is not a good marker. ‘By evaluating the change in serum ferritin after statin treatment, we attempted to determine whether statins exert an anti-inflammatory effect in these patients and whether this effect was correlated with lowered resistance to ESAs.
  2. Line 229: ‘Moreover, ferritin levels tended to non-significantly decrease after statin use, indicating a decrease in the inflammatory index’ the sentence should be rechecked that I mentioned above.
  3. Line 238: Should be rechecked for the same reason about the ferritin.

Response: Thank you for your valuable remarks. As we mentioned above, the concern of representative of ferritin as an inflammatory marker may lowered owing to the inclusion of both CKD and ESKD patients in our study.

Reviewer 2 Report

In the present systematic review and meta-analysis, Tsai et al. assessed the possible pleiotropic effect of erythropoietin-stimulating agents involving 10 randomized control trial/cohort study/pre-post observational study of ESKD patients receiving hemodialysis and regular erythropoietin-stimulating agents. The Authors found that Hb levels were higher in the study group than in the control group, both at baseline and at the endpoint; and an increase in ferritin levels in the Stain treated patients compared with controls. They concluded that statin may improve renal anemia in ESKD patients receiving hemodialysis and regular erythropoietin-stimulating agents, suggesting the need for future studies with more rigorous methodology and larger sample size study.

The issue is a new one (to my knowledge, it is the first meta-analysis about this issue).
The introduction provide an exhaustive background.

The methodological approach is appropriate, with the properly assessing of risk of bias in randomized controlled trials.

The analysis doesn’t take into account the presence of diabetes, which have a dramatic impact on ESKD.

In the Discussion section, the message is clearly drive to the reader.

The conclusions correlate to the results found but have to be down-toned. As the Authors themselves explain in the text, there are several important limitations, as the relatively small sample sizes of the studies involved, the most of included studies were observational studies and single-arm studies.

Author Response

Reviewer 2

Dear reviewer:

We are grateful for the time and effort you and each of the editors have dedicated to provide insightful feedback on ways to strengthen our paper. It is a novel idea and indeed the first meta-analysis in this issue. As your concern, Diabetes Mellitus (DM) patients did play an important role in the causes of chronic kidney disease (CKD). In our meta-analysis, all of the selected studies included patients with DM. In addition, we did not exclude diabetic CKD patients. Therefore, we could not conduct a subgroup analysis based on CKD underlying etiologies. We hope that our response satisfactorily addresses the issues and concerns you have noted.

Reviewer 3 Report

Anemia in CKD and hemodialysis is an actual subject and as is mentioned up to 10% of the patients still don't meet efficiency criteria. You test a hypothesis of efficiency of the statins in controlling inflammation and so improving Hb, ferritin parameters. 

The metanalysis steps and procedures are correct regarding methodological point of view, statistical analysis is Ok and in line with available data.

In the discussion chapter you address the correct questions and there are well presented missing data in selected studies.

Conclusions are clear and present possible efficiency of the statins based on available data, even as they are non-statistically significant.

Also, recommendation for new well-designed studies and with sufficient subjects are OK.

Author Response

Reviewer 3

Dear reviewer:

We are grateful for the time and effort you and each of the editors have dedicated to provide insightful feedback on ways to strengthen our paper. We had deliberately and carefully revised our manuscript according to the reviewer’s valuable opinions. We also hope that our edits and the responses we provide satisfactorily address all the issues and concerns you and the reviewers have noted.

Reviewer 4 Report

The authors presented a study addressing an important topic in nephrology: the treatment of renal anemia. They conducted a meta-analysis of available studies, both RCTs and observational studies, to answer important clinical questions. The manuscript needs the corrections listed below before publication:

1. Typo in the abstract: "Stain, an inhibitor of HMG-CoA (...)".

2. Abstract and Methods: "We searched the PubMed, Embase, Medline...". The MEDLINE database is a part of PubMed so an additional search in MEDLINE via another site (OVID?) was unnecessary. How did the authors access MEDLINE?

3. Methods: "This systematic review and meta-analysis was conducted following the preferred reporting items for systematic reviews and meta-analyses (PRISMA) 2020 statement". PRISMA 2020 is guidelines for reporting, not conducting, systematic reviews. The sentence should be rewritten: it was reported following (...).

4. Provide the value of the correlation coefficient used to determine the missing SD.

5. The authors should check whether they provided the correct version of R (they reported 1.2.5033, while the current one is > 4.0). Moreover, they did not specify the R package used to perform a meta-analysis.

6. The Authors wrote: "Meta-analyses were performed (...) with random effects model". However, Figure 1 and Fig. 2 provided data based on the fixed-effect model.

7. The "Study selection" section has to be before "Study and patient characteristics".

8. The authors correctly used "ESKD" in the manuscript, whereas "ESRD" (old term) in Table 1. It should be corrected.

9. Separate analysis of the Hb concentration at baseline (Fig. 5) is meaningless in a meta-analysis of RCTs. This outcome can be assessed by either analysis of the MD at the endpoint (Fig. 6) or MD of change-from-baseline (Fig. 10). Please group analyses of one outcome in one place (paragraph/chapter).

10. Again, a separate analysis of the ferritin concentration at baseline (Fig. 7; the title of this figure is incorrect) is meaningless in a meta-analysis of RCTs. This outcome can be assessed by either analysis of the MD at the endpoint (Fig. 8) or MD of change-from-baseline (not provided by the authors).

11. Please provide units in all analyses (e.g. g/dL).

12. Results of the change in ERI should be provided in the Results section (they are in the Discussion section). If they cannot be meta-analyzed, you can tabulate or summarize them.

Author Response

Reviewer 4

  1. Typo in the abstract: "Stain, an inhibitor of HMG-CoA (...)".

Response: Thank you for your reminder. We have corrected the error in our abstract.

  1. Abstract and Methods: "We searched the PubMed, Embase, Medline...". The MEDLINE database is a part of PubMed so an additional search in MEDLINE via another site (OVID?) was unnecessary. How did the authors access MEDLINE?

Response: Thank you for your valuable reminder. We did not access Medline via another site. We have deleted the term “Medline” in the abstract and method.

  1. Methods: "This systematic review and meta-analysis was conducted following the preferred reporting items for systematic reviews and meta-analyses (PRISMA) 2020 statement". PRISMA 2020 is guidelines for reporting, not conducting, systematic reviews. The sentence should be rewritten: it was reported following(...).

Response: Thank you for your valuable comments. We have rewritten the sentence.

  1. Provide the value of the correlation coefficient used to determine the missing SD.

Response: Thank you for your reminder. The values of correlation coefficient of the statin group and control group used for calculating missing SD are 0.96715 and 0.76565, respectively.

  1. The authors should check whether they provided the correct version of R (they reported 1.2.5033, while the current one is > 4.0). Moreover, they did not specify the R package used to perform a meta-analysis.

Response: Thank you for your reminder. We have rechecked and revised the correct version of R. In addition, the R packages used for performing meta-analysis are added.

  1. The Authors wrote: "Meta-analyses were performed (...) with random effects model". However, Figure 1 and Fig. 2 provided data based on the fixed-effect model.

Response: Thank you for your valuable reminder. We apologized for providing the erroneous result preformed with fixed-effect model. We have corrected Figure 1, Figure 2, and the values written in the passage. The incremental tendency of Hb after statin use remained the same.

  1. The "Study selection" section has to be before "Study and patient characteristics".

Response: Thank you for your valuable remarks. We have changed the sequence of paragraphs.

  1. The authors correctly used "ESKD" in the manuscript, whereas "ESRD" (old term) in Table 1. It should be corrected.

Response: Thank you for your reminder. We have revised the term “ESRD” to “ESKD” in Table 1.

  1. Separate analysis of the Hb concentration at baseline (Fig. 5) is meaningless in a meta-analysis of RCTs. This outcome can be assessed by either analysis of the MD at the endpoint (Fig. 6) or MD of change-from-baseline (Fig. 10). Please group analyses of one outcome in one place (paragraph/chapter).
  2. Again, a separate analysis of the ferritin concentration at baseline (Fig. 7; the title of this figure is incorrect) is meaningless in a meta-analysis of RCTs. This outcome can be assessed by either analysis of the MD at the endpoint (Fig. 8) or MD of change-from-baseline (not provided by the authors).

Response: Thank you for your insightful comments. The main point of this analysis was to provide a reference to those who may be interested in the baseline value comparison. In addition, it could highlight the trend of variation of our outcomes, especially in ferritin since we could not calculate the missing SD of its change-from-baseline. We would not emphasize the result of this analysis.

  1. Please provide units in all analyses (e.g. g/dL).

Response: Thank you for your valuable reminder. We have added the units in all analyses.

  1. Results of the change in ERI should be provided in the Results section (they are in the Discussion section). If they cannot be meta-analyzed, you can tabulate or summarize them.

Response: Thank you for your valuable reminder. We have added a paragraph summarizing the result of the change in ERI in our result section.

Round 2

Reviewer 4 Report

The article has been greatly improved. Some minor issues:

1. There is no such R package as "metaphor". It is metafor (https://www.metafor-project.org/doku.php/metafor).

2. There are very high differences in the baseline ERI reported for N Nand vs other studies. Are all studies using the same units? Please add units.

Author Response

Response to Comments

Reviewer 4

  1. There is no such R package as "metaphor". It is metafor (https://www.metafor-project.org/doku.php/metafor).

Response: Thank you for your valuable reminder. We have corrected the mistake.

  1. There are very high differences in the baseline ERI reported for N Nand vs other studies. Are all studies using the same units? Please add units.

Response: Thank you for your valuable comments. All of the studies mentioning ERI use the same units: IU/kg/week/g/dL. We have added the units. In addition, we did notice the highly differences of outcomes reported from N Nand compared with other studies. However, as we have mentioned in our discussion, neither the authors of N Nand tabulated the demographic data of the experimental and control groups nor discussed whether the two groups were comparable, which obstructed us to identify the heterogeneity source.